# Characterizing and predicting person-specific, day-to-day, fluctuations in walking behavior

**Guillaume Chevance**[1,2,3]*, **Dario Baretta**[4], **Matti Heino**[5], **Olga Perski**[6], **Merlijn Olthof**[7], **Predrag Klasnja**[8], **Eric Hekler**[2,3], **Job Godino**[2,3]

**1** ISGlobal, Barcelona Institute for Global Health, Barcelona, Spain, **2** Center for Wireless & Population Health Systems, The Qualcomm Institute and the Department of Family Medicine and Public Health, University of California, San Diego, San Diego, CA, United States of America, **3** Exercise and Physical Activity Resource Center, University of California, San Diego, San Diego, CA, United States of America, **4** Independent Researcher, PhD in Psychology, Geneva, Switzerland, **5** Faculty of Social Sciences, University of Helsinki, Helsinki, Finland, **6** Department of Behavioural Science and Health, University College London, London, United Kingdom, **7** Behavioural Science Institute, Radboud University, Nijmegen, The Netherlands, **8** School of Information, University of Michigan, Ann Arbor, MI, United States of America

* guillaume.chevance@isglobal.org

**Data Availability Statement:** The data and code for the statistical analyses used in the present study are available on Open Science Framework, https://osf.io/64cmv/.

## Abstract

Despite the positive health effect of physical activity, one third of the world's population is estimated to be insufficiently active. Prior research has mainly investigated physical activity on an aggregate level over short periods of time, e.g., during 3 to 7 days at baseline and a few months later, post-intervention. To develop effective interventions, we need a better understanding of the temporal dynamics of physical activity. We proposed here an approach to studying walking behavior at "high-resolution" and by capturing the idiographic and day-to-day changes in walking behavior. We analyzed daily step count among 151 young adults with overweight or obesity who had worn an accelerometer for an average of 226 days (~25,000 observations). We then used a recursive partitioning algorithm to characterize patterns of change, here sudden behavioral gains and losses, over the course of the study. These behavioral gains or losses were defined as a 30% increase or reduction in steps relative to each participants' median level of steps lasting at least 7 days. After the identification of gains and losses, fluctuation intensity in steps from each participant's individual time series was computed with a dynamic complexity algorithm to identify potential early warning signals of sudden gains or losses. Results revealed that walking behavior change exhibits discontinuous changes that can be described as sudden gains and losses. On average, participants experienced six sudden gains or losses over the study. We also observed a significant and positive association between critical fluctuations in walking behavior, a form of early warning signals, and the subsequent occurrence of sudden behavioral losses in the next days. Altogether, this study suggests that walking behavior could be well understood under a dynamic paradigm. Results also provide support for the development of "just-in-time adaptive" behavioral interventions based on the detection of early warning signals for sudden behavioral losses.

**Funding:** The present research was funded by a grant from the National Heart, Lung, and Blood Institute (NHLBI); R01HL136769 awarded to the last author (JG). The first author (GC) acknowledges support from the Spanish Ministry of Science and Innovation and State Research Agency through the "Centro de Excelencia Severo Ochoa 2019-2023" Program (CEX2018-000806-S), and support from the Generalitat de Catalunya through the CERCA Program.

**Competing interests:** The authors have read the journal's policy, and the authors of the study have the following competing interests to declare: Accenture provided support in the form of a salary for DB. This does not alter our adherence to PLOS ONE policies on sharing data and materials. There are no patents, products in development or marketed products associated with this research to declare.

## Introduction

It is widely recognized that physical activity directly protects against the development and exacerbation of non-communicable diseases and mental health issues, and improves quality of life [1]. Physical activity is also, in some forms such as active transportation, a key contributor to climate change mitigation and air pollution reduction, which indirectly protect population health [2]. Despite these positive effects, one third of the world's population is estimated to be insufficiently active and, so far, the promotion of physical activity globally has been described as "*largely unsuccessful*" [3]. Although several explanations for this relative failure have been proposed (see for example [4, 5]), little prior research has been conducted to understand the day-to-day, "high-resolution", and long-term changes in physical activity behavior, despite clear theoretical and practical benefits to do so. Indeed, gaining a better understanding of the resolution of change could provide important information to refine current physical activity guidelines and theories to reflect a more dynamic perspective and, ultimately, inform new forms of quasi-real time physical activity interventions [6–9]. The present study aimed to improve our understanding of day-to-day changes in physical activity within individuals over several months. Notably, this study focuses on walking behavior, a central component of physical activity and an accessible, inexpensive and low-impact means for individuals to meet national and international physical activity guidelines [10, 11].

### Physical activity behavior changes

Evaluation of physical activity interventions typically incorporate measurement bursts, i.e., regular measurement of physical activity for 3 to 7 days, that are dispersed across time, such as immediately before taking part in an intervention, immediately after an "intervention phase", and, occasionally, a few weeks or months after the intervention is completed (e.g., [12]). Using this measurement approach, physical activity levels within each measurement burst are often aggregated at the weekly level (e.g., minutes/week of moderate-to-vigorous intensity physical activity), and insight into the high-resolution day-to-day variations in physical activity removed through aggregation [9]. As both a cause and consequence to this methodological paradigm, physical activity changes have been mostly conceptualized as a relatively linear and deterministic process (e.g., "the *adoption and maintenance* paradigm"; [13–16]). Nonetheless, recent studies adopting continuous measures of physical activity via accelerometry and investigating the within-person variability in that behavior have shown that physical activity levels can vary extensively within individuals over time and across different contexts [17–20]. To date, no studies have characterized the within-person patterns of change in physical activity over several months; however, recent findings speak to the possibility that physical activity fluctuates in a potentially strong irregular fashion at the individual level [17–20].

In this regard, complex systems theory offers a unique framework to characterize and understand dynamics and discontinuous/irregular patterns of change in human behavior [8, 16, 21, 22]. According to this theoretical framework, complex systems can experience abrupt and discontinuous changes from one stable state to another over time (e.g., [23]). These changes are sometimes referred to as *abrupt shifts*, *order transitions* or *sudden gains and losses* ([24]; we will henceforth refer to them using this last term). Numerous examples of sudden gains and losses in complex systems exist in the literature, from asthma attacks [25], shifts in motivational flow [26], improvements or aggravations of symptoms in individuals with depression, mood and/or anxiety disorders [27], and, at a broader level, forms of societal collapse [28] or climatic change [29]. In the health behavioral science literature, recent studies found that sleep stage transitions [30], as well as sedentary behaviors (i.e., time spent sitting) [31], were characterized by such irregular patterns of change. More volitional behaviors, such

as daily changes in alcohol consumption, have also been successfully modelled with methods allowing the consideration of non-linearity and non-stationarity [32]. Although there have been prior and recent calls to study the dynamics of physical activity behavior changes through the lens of complex systems theory [8, 16], empirical investigations are lacking for this behavior.

## Early warning signals for sudden behavioral changes

Beyond characterizing the dynamic nature of behavior change processes, early findings suggest that sudden gains and losses in complex systems can sometimes be predicted via *early warning signals* [24, 33]. One kind of signal corresponds to the presence of critical fluctuations in the behavior, as observed within time series data from a given system (e.g., an individual), which can be interpreted as an indicator of an imminent sudden gain or loss [34]. For example, research conducted within the psychotherapy setting showed that sudden gains and losses in symptom severity in individuals diagnosed with a mood disorder could be predicted by critical fluctuations in measures of the therapeutic process a few days before each transition [35]. This study used scores of *dynamic complexity* to quantify the amount of fluctuation [34]. The authors reported a positive and significant association between different dynamic complexity indicators and subsequent occurrence of sudden gains and losses [35]. Notably, this study found that greater fluctuations in measures of the therapeutic process were prospectively associated with the occurrence of sudden gains and losses in the next few days (see Fig 1 for a conceptual illustration of this process). The detection of behavioral fluctuations, quantified by indicators of dynamic complexity, thus represents a potentially useful way for clinicians and

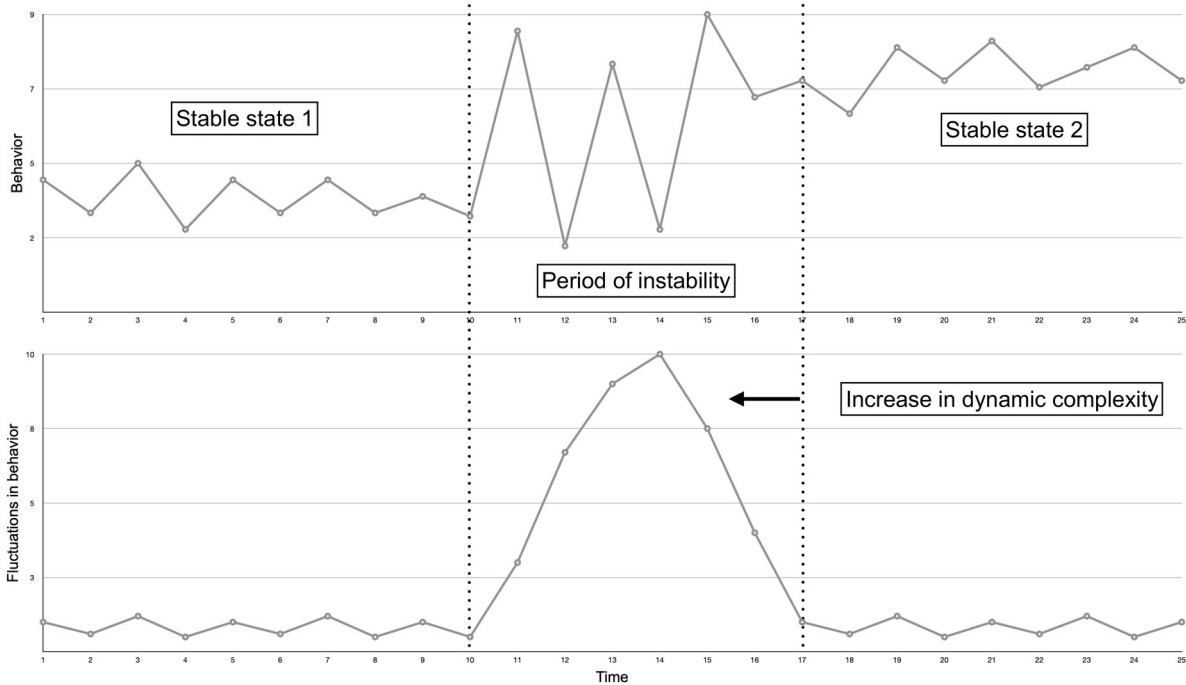

**Fig 1. Conceptual illustration of critical fluctuations in a given behavior followed by a sudden gain (upper panel), expressed as the level of dynamic complexity (lower panel).** The transition from a stable state (starting at day 1 and ending around day 10) to a second stable state (starting around day 17) is characterized by a period of instability in which the behavior of the system displays critical fluctuations (upper panel). The greater the fluctuations in the behavior, the higher the dynamic complexity score (lower panel). Work in critical transitions across various systems indicate that instability can be observed shortly before a phase transition; increases in dynamic complexity can thus be seen as a form of early warning signal for either sudden gains and/or losses in a given system [35, 37]. Note that the Figure's x axis represents continuous time but does not refer to a specific time unit, as this approach can be used to describe both rapid- and slow-evolving processes. The time unit here can thus conceptually vary from seconds to years.

intervention designers to anticipate sudden behavioral gains and losses before they occur [24]. Applied to physical activity, the early prediction of imminent behavioral transitions represents a potentially valuable signal for delivering interventions "just-in-time" [36].

It is theoretically plausible that physical activity behavior is usefully described from a complex systems perspective; however, there is a paucity of empirical work to test this assumption. If empirical work suggests that there are inherent sudden gains and losses, and potential complexity, in physical activity changes within individuals over time, then that would have important implications for physical activity research methods, theories, and interventions. If strong discontinuous patterns of change are observed, then the current measurement approach, which largely relies on snapshots of behavior across time, would be demonstrably insufficient to understand behavioral dynamics. Consequently, there would be a need to move from low- to high-resolution physical activity assessments [9]. If results illustrate discontinuous patterns of change, this would also establish the need for behavioral theories of physical activity that better account for the dynamic, discontinuous and potentially complex nature of physical activity [6, 7]. Finally, in regard to interventions, if the sudden gains and losses exist, the capacity to identify early warning signals for these state shifts in physical activity would be highly valuable for realizing and optimizing just-in-time adaptive interventions [36]. In particular, critical fluctuations, characterized by an increase in dynamic complexity, could be used to identify states of vulnerability, opportunity, or "teachable moments" for individuals (i.e., periods in which individuals would benefit from support if they are receptive to it [38, 39]).

### The present study

This study aimed to (*i*) characterize within-person, day-to-day patterns of change in physical activity (i.e., daily step count); and (*ii*) examine whether critical fluctuations in walking behavior, expressed as a dynamic complexity score, were associated with the subsequent occurrence of a sudden gain or loss in the following days. Based on previous literature in psychology and behavioral sciences, we expected to observe discontinuous patterns of change in walking behavior within individuals, including sudden gains and losses preceded and followed by relatively stable behavioral states (see Fig 1, upper panel). We also expected heterogeneity in state transitions between participants, with participants experiencing either few or many sudden gains and losses. Second, we expected a significant and positive association between dynamic complexity (derived from the observation of fluctuations in walking behavior) and the subsequent occurrence of a sudden gain or loss in the following days (Fig 1, lower panel).

## Materials and methods

To test these hypotheses, we used data from an ongoing two-year, three-arm, parallel-group randomized controlled trial (RCT) targeting weight loss among young adults with overweight or obesity (ClinicalTrials.gov Identifier: NCT03907462). Only the participants recruited for this RCT from April 16[th] 2019 (i.e., the beginning of the trial) to March 1[st] 2020 (i.e., before Covid-19 lockdown measures expected to influence physical activity were introduced in San Diego, California) were included in the present study. All relevant ethical regulations have been followed and the ethical committee of the University of California San Diego approved the study protocol. Participants have completed informed consent.

### Participants and procedure

The analytic sample comprises only the participants from the RCT who had worn an activity monitor for at least 90 days (i.e., 3 months) with less than 20% of missing daily observations (i.e., inclusion criteria for the present study). Complete observations were defined as at least

600 minutes of valid minute heart rate signal per day (i.e., wear time indicator). Inclusion criteria for the RCT included being aged between 18 and 35 years, being affiliated with one of 3 major universities in San Diego (CA) as a student, faculty, staff member, or alumni and being overweight or obese ($25 > = BMI < 40$ kg/m$^2$). Participants with comorbidities such as psychiatric and/or medical conditions were not eligible for the RCT.

The RCT is a 24-month study including (*i*) a group with activity monitors and connected digital scales, text messaging and lifestyle interventions based on social media, targeting weight loss through behavioral changes, (*ii*) a group with the abovementioned intervention package and additional phone-based health coaching sessions and (*iii*) a control group with activity monitors and digital scales alone. As the trial is still ongoing, group assignment remained unknown for the present study, and was not modelled in our analyses.

## Measures

Data on walking behavior (i.e., daily step count) were collected with a Fitbit Charge 3 (Fitbit Inc, San Francisco, CA, USA). Participants were asked to wear the wrist-worn device continuously during the trial according to manufacturer recommendations (i.e., to wear the device on one's non-dominant arm and up to three finger widths above the wrist bone).

Steps are estimated from the Fitbit via triaxial accelerometry measuring gravitational acceleration in the anterior-posterior [x], cranial-caudal [y], and medial-lateral [z] planes. It should be noted that this indicator of physical activity (i.e., step count) was chosen over other metrics offered by the device (e.g., physical activity intensity) because the estimation of steps from Fitbit sensors range from good to excellent in comparison with direct observation [40], while the validity of other metrics remains questionable [41]. With regards of descriptive statistics, weight and height were measured objectively in the lab and used to compute participant's BMI in kg/m$^2$. Age and gender were self-reported.

## Data analysis

**Time series characteristics.** The mean length of the time series across participants was 226 days and the mean number of missing days per participant was 20 days (range = 0–61 days). Missing days were imputed using the Kalman Filter method (i.e., a procedure to compute the likelihood of a time series which is the outcome of a stationary autoregressive moving average process or non-stationary autoregressive integrated moving average process; see for further details [42]). This method, recommended for univariate time series imputation [43], was preferred over other techniques (e.g., 'last observation carried forward') based on visual inspections of the time series.

**Characterization of gains and losses.** Following the approach applied by Olthof et al. (2020), we used a recursive partitioning algorithm to characterize within-person behavioral changes over the course of the study using regression trees from the package *rpart* [44] for the R software environment [45]. Regression trees enable to identify nodes and partition a continuous variable y in recursive splits based on different predictors. In the present study, we partitioned the variable steps based on the progressive day of the intervention as predictor. Once we identified the splits for the variable steps for each participant separately, we classified them as gain or losses if they met additional criteria defined on the basis of previous research. Specifically, sudden gains and losses were defined as a shift toward a lower (loss) or higher (gain) number of steps over time, provided that two conditions (i.e., criteria) were simultaneously met:

1. Based on past research from our group [46–48], gains or losses for each participant were defined as a 30% increase or reduction in steps relative to the median number of steps, as

assessed over the study period. For example, for a participant with a median of 7000 steps/day, a sudden gain was defined as an increase greater than 2100 steps per day (+30%) compared with the median, while a sudden loss was defined as a drop greater than 2100 steps per day (-30%) compared with the median. We used the median number of steps instead of the mean to manage potential skewed step count distributions at the individual level.

2. In addition to these +/- 30% criteria, and to control for known differences in walking behavior between weekends and weekdays [49], sudden gains and losses had to be observed for stable periods of at least 7 days during which no further shifts in step count occurred. In other words, these changes of +/- 30% had to last at least 7 days, otherwise they were not classified as sudden gains or losses. Consequently, periods of high variability in steps without a stable period of at least 7 days (i.e., changes greater than +/- 30% from one day to another) were not classified as sudden gains or losses with our algorithm.

**Local dynamic complexity calculation.**  After the identification of gains and losses, dynamic complexity was estimated from the fluctuations in the steps times series. We used for this purpose an algorithm designed specifically to identify critical fluctuations in time series and implemented with the R package *casnet* [50].

The dynamic complexity score is the product of a fluctuation measure, *F*, and a distribution parameter, *D*, for each unit of the time series (here, each day). The fluctuation measure, *F*, is sensitive to the amplitude and frequency of change in the time series, with *F* being at its maximum when the dynamics of the data vary from one observation to another with large (e.g., between the minimum and maximum values of the outcome) and regular frequency. The distribution parameter, *D*, measures the deviance of the data points from the range of possible values within the time series. This parameter increases when irregular changes from one observation to another occur (see [34]).

The two measures, *F* and *D*, were computed from moving time windows over the course of the time series. Based on previous literature [34, 35], we used a backward 7-day overlapping window, which means that *F* and *D* for a particular day *n* was computed on the basis of fluctuations in walking behavior in the past 7 days. In other words, the scores for day *n* was computed based on the aggregation fluctuation intensity in the past 7 days, between day *n*-7 and day *n*.

The dynamic complexity score (*F* x *D*) was then used to derive a continuous predictor of sudden gains and losses called *local dynamic complexity*. This score reflects the highest complexity score (*F* x *D*) in the 3 days preceding a possible gain or loss (as conceptually illustrated in Fig 1, lower panel, day #14). This 3-day time window was chosen based on previous literature [35] and because, theoretically, critical fluctuations are likely to appear a few days before a sudden gain or loss (although please see the sensitivity analyses in the Results section with varying time windows).

For mathematical details, see the document entitled "Dynamic Complexity" in the supplemental material available at https://osf.io/64cmv/).

**Associations between local dynamic complexity and sudden gains and losses.**  The association between local dynamic complexity (i.e., the independent variable) and the occurrence of sudden gains and losses (i.e., the dependent variable) was tested with discrete-time multilevel event-history analysis through a generalized linear mixed-effects model. Individual differences in the number of gains and losses were controlled by including a random intercept. Adopting a "maximal approach" [51], random slopes were also included for the two continuous independent variables, time and local dynamic complexity. The equation, in the language

of the package *lme4* [52], is provided in the equation below:

$$Sudden\ gains\ or\ losses \sim Time + Duration + Local\ Dynamic\ Complexity$$
$$+ (1 + Time + Duration + Local\ Dynamic\ Complexity \mid\mid Participant\ ID)$$

As recommended for this type of analysis [53], the progressive number of days from the beginning of the trial was included as a control variable (e.g., the likelihood of sudden gains and losses may be higher at the beginning of the study compared with the end). Further, the effect of time between events (e.g. sudden gains, losses and the start of data collection) was modeled by including a "duration indicator", i.e. the number of days passed between events. The inclusion of this duration indicator is essential in event-history analyses to (*i*) control for censored cases (i.e., participants who did not experience a sudden gain or loss within the observation period) and (*ii*) reliably estimate cases with multiple sudden gains or losses [53]. Similar models were then conducted separately for the occurrence of gains (1 = occurrence of a sudden gain; 0 = no sudden gain) and losses (1 = occurrence of a sudden loss; 0 = no sudden loss). Odds ratios of the fixed effects coefficients were estimated with 95% likelihood profile confidence intervals.

**Code and data availability.** Statistical analyses were performed in R version 4.0.2. The data and code for the statistical analyses used in the present study are available on Open Science Framework, https://osf.io/64cmv/.

## Results

### Participant and time series characteristics

The final sample includes 151 young adults with overweight or obesity [57% Female; Mean age = 23 years (range = 18–38 years), mean BMI at baseline = 30 kg/m$^2$ (range = 25–44 kg/m$^2$)], who had worn an activity monitor measuring their daily step count for an average of 226 days (range = 107–320 days). The mean number of steps/days during the study across participants was 9687 steps/day (range = 508–53131 steps/day, median = 9110).

### Objective 1: Characterization of sudden gains and losses

Participants experienced an average of 6 gains or losses during the study (range = 0–14), with an average of 3 gains (i.e., +30% compared with median steps for a period of at least 7 days) and 3 losses (i.e., -30% compared with median steps for a period of at least 7 days) per participant. Ten participants only experienced gains, six participants only experienced losses and one participant did not experience any gains or losses according to our criteria. The average period of time between two gains or losses was 31 days (range = 9–110 days). The average change in steps observed when a gain or loss occurred was 4900 steps (5022 steps for gains and 4784 steps for losses).

To illustrate this, Fig 2 presents the time series for six participants. The figure includes: the participant experiencing the highest number of gains and losses (participant #4); the participant experiencing no significant gains or losses (participant #100); one participant presenting only one loss (participant #121) and one participant presenting only one gain (participant #60); and finally, two participants that are representative of the total sample, experiencing a relatively equal number of gains and losses (participants #48 and #79). Similar visualizations for the 151 participants' time series are available in the S1 Fig.

### Objective 2: Associations between local dynamic complexity scores and sudden gains and losses

The association between local dynamic complexity scores and the occurrence of sudden gains and losses was then tested with generalized linear mixed-effects models. The first model

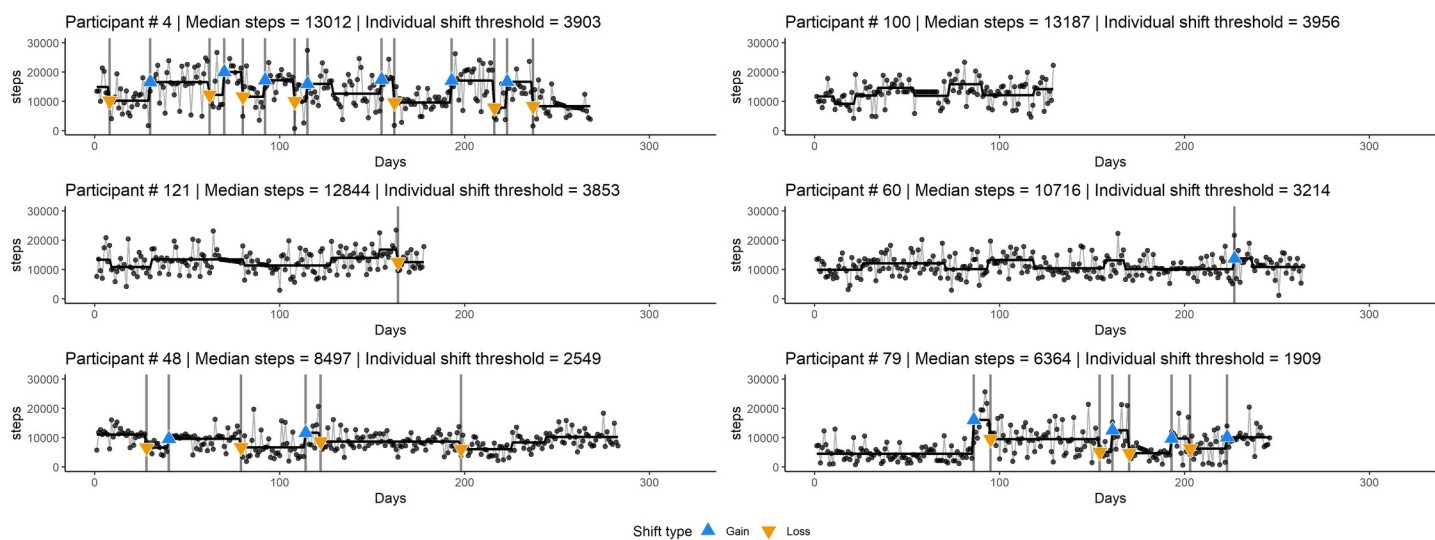

**Fig 2. Time series for six participants with heterogeneous patterns of change in walking behavior.** Some apparent gains and losses were not classified as "significant" (as indicated by the blue and orange triangles) because they were either smaller than 30% of the median or not preceded and followed by a stable 7-day period.

included the occurrence of a sudden gain or loss as a binary outcome (1 = occurrence of a gain or loss; 0 = no gain or loss) and the local dynamic complexity score as the predictor of interest. The same model was then used to test the associations between local dynamic complexity and gains and losses separately in order to examine potential differences when the outcome was framed as a gain rather than a loss, and vice versa.

Results are presented in Table 1. Local dynamic complexity positively predicted subsequent behavioral shifts with an odds ratio (*OR*) of 1.14, 95%CI [1.05, 1.24], indicating that an increase in local dynamic complexity of 1 *SD* relates to 14% increased odds of either a gain or a loss in the next 3 days. This result was driven by a positive association between local dynamic complexity score and sudden losses, *OR* = 1.44, 95%CI [1.28, 1.58]. Indeed, an increase in local dynamic complexity of 1 *SD* relates to 44% increased odds of experiencing a transition to lower physical activity levels. In contrast, local dynamic complexity tended to be negatively related (although not statistically significant) with behavioral gains, *OR* = 0.88, 95%CI [0.76, 1.00]. Time (i.e., the progressive number of days from the start of data collection) was negatively associated with the occurrence of both types of behavioral shifts, indicating that sudden gains and losses were less likely to occur at the end of the data collection period compared with the beginning. In contrast, duration (i.e., the number of days between events) was positively associated with the occurrence of both types of behavioral shifts, indicating that the longer the time since the beginning of the study or a prior gain or loss, the more likely it is that a participant will experience a sudden gain or loss.

**Table 1. Associations between local dynamic complexity scores and sudden gains and losses in walking behavior.**

| Predictors | Gains and losses (N = 151; k = 26882) | | Gains only (N = 144; k = 25924) | | Losses only (N = 140; k = 25031) | |
|---|---|---|---|---|---|---|
| | *OR* | 95% CI | *OR* | 95% CI | *OR* | 95% CI |
| Time | 0.85 | [0.77, 0.93]* | 0.70 | [0.61, 0.79]* | 0.60 | [0.51, 0.71]* |
| Duration | 1.21 | [1.12, 1.32]* | 2.06 | [1.84, 2.30]* | 2.11 | [1.86, 2.40]* |
| LDC | 1.14 | [1.05, 1.24]* | 0.89 | [0.77, 1.01] | 1.43 | [1.28, 1.58]* |

*OR* = odds ratio; CI = confidence intervals; LDC = local dynamic complexity.

* = statistical significance based on confidence intervals.

To summarize, results show that higher local dynamic complexity (or higher fluctuations in one's daily step count) are positively associated with the odds of observing a sudden loss of at least 30% (compared with one's median step count) in the next three days, lasting for a period of at least a week. However, fluctuations in one's daily step count is not significantly associated with the odds of observing a sudden gain of at least 30% compared with one's usual step count in the next three days, lasting for at least a week (see Table 1).

## Sensitivity analyses

Sensitivity analyses were conducted to test the robustness of the results for varying levels of gains and losses (i.e., +/-20% and +/-40% of the median number of steps, instead of +/-30%, as specified in the main analyses), and different time-lags for the local dynamic complexity score (i.e., higher scores in the past two and four days, instead of the past three days, as specified in the main analyses). Results of the sensitivity analyses were compared with regards to their effects on the identification of gains and losses (objective 1) and the association between local dynamic complexity and the occurrence of sudden gains and losses (objective 2).

As shown in Table 2, lower (+/-20%) and higher (+/-40%) thresholds lead respectively to the identification of more (N = 1180) and less (N = 595) sudden gains and losses in comparison with the main analyses (+/-30%; N = 862). Using a more conservative threshold (+/-40% of the median number of steps) resulted in the detection of a lower number of sudden gains and losses, while using a less conservative threshold (+/-20% of the median number of steps) resulted in the detection of a higher number of events.

The associations between local dynamic complexity and the occurrence of sudden gains and losses were substantially unchanged (notably the prediction of sudden losses) in the sensitivity analyses with different time windows chosen for the computation of the local dynamic complexity score. Results from these 3 (thresholds for the identification of gains and losses) x 3 (time-lapses for the computation of local dynamic complexity) x 3 (gains and losses separately and together) sensitivity analyses are presented in the S1 Table.

## Discussion

Findings from this study provide new empirical insights into the idiographic, day-to-day, changes in walking behavior over time in participants of a weight loss program. Regarding our first objective, the results revealed that, when analyzed at the daily-level, walking behavior exhibits discontinuous changes that can be described as sudden gains and losses, as conceptualized in complexity theory. Indeed, on average participants to this study experienced six sudden gains and losses of at least +/-30% of their individual steps level over a mean period of 226 days. Results also show heterogeneity in the patterns of change in walking behavior, with participants experiencing between 0 and up to 14 behavioral gains or losses during the study period. Regarding our second objective, significant associations between critical fluctuations in walking behavior, characterized here as increases in local dynamic complexity, and the

**Table 2. Effect of different thresholds on the identification of behavioral gains and losses.**

| Thresholds | Gains and losses | Gains only | Losses only |
|:---:|:---:|:---:|:---:|
| **20%** | N = 1180 | N = 563 | N = 617 |
| **30%** | N = 862 | N = 422 | N = 440 |
| **40%** | N = 595 | N = 295 | N = 300 |

**Note**. Thresholds are estimated based on the median number of steps during the study period for each participant.

subsequent occurrence of sudden gains and losses in the next three days were observed. Notably, increased local dynamic complexity was predictive of increased odds of sudden losses. The odds of having a sudden loss was 43% higher after an increase in local dynamic complexity. Overall, these findings provide early empirical support of our hypotheses that walking behavior is subject to irregular, day-to-day changes and that sudden losses can be anticipated with some accuracy from the analyses of critical fluctuations computed from physical activity behavior two to four days previous.

The first set of results (i.e., the identification of sudden gains and losses), is in accordance with both (*i*) previous empirical results, showing that psychological, physiological, social and behavioral systems, when observed at a high-resolution, might changes in a discontinuous fashion (e.g., [26, 32]), and (*ii*) complex systems theory assumptions, which postulate that any natural system is "bubbling" with change and tends to evolve in a non-stable manner [54]. Our observations thus provide support for several recent position papers arguing that changes in health behaviors can be fruitfully studied with research designs, methods, and statistics that reflect these behavioral dynamics [8, 9, 55]. Further, at the theoretical level, results from this study support the adoption of complex systems theory to better understand the dynamics of health behaviors over time and across contexts [6, 16]. With regards to walking behavior specifically, visual inspection of the time-series (see S1 Fig) also provides support to previous research (conducted at the group level and over short and aggregated periods of time) showing that significant variance in physical activity data could be observed at the intra-individual level, thus varying extensively not only between individuals but also within individuals [20].

The second set of results (i.e., associations between local dynamic complexity and sudden behavioral shifts), are partially aligned with previous literature. We observed a significant and positive association between our marker of critical fluctuations (i.e., local dynamic complexity) and the occurrence of sudden behavioral shifts in the following days. This confirmed assumptions from complex systems theory that changes within a system tends to be characterized by increased variability in system behavior before reorganization [34] (see Fig 1 for an illustration). However, this result was driven by a relatively strong positive association between critical fluctuations and the occurrence of a sudden behavioral loss. In contrast, although not significant, greater fluctuations were related to reduced odds of behavioral gains. Previous empirical work, conducted by Olthof et al. (2019), also found differences in the prediction of improvement (sudden losses) and deterioration (sudden gains) in psychological symptoms: (*i*) a positive and significant association between higher fluctuations and the occurrence of sudden symptoms improvements in psychotherapy and (*ii*) a positive but not-significant association between higher fluctuations and the occurrence of sudden symptoms deteriorations [35]. However, these analyses were explorative and possibly under-powered [35]. It is possible that different indicators of critical fluctuations have different predictive values for gains and losses depending on contextual parameters (i.e., weight loss program *versus* psychotherapy) or according to the outcome specificity (i.e., walking behavior *versus* depressive symptoms). Whether different kinds of gains or losses can be predicted from different critical fluctuations indicators remains an open question for future research inside and outside the field of physical activity. In any case, the fact that different associations are observed in the literature between indicators of dynamic complexity and sudden gains and losses invite to analyze these two kinds of behavioral shifts distinctively instead of pulled together.

Finally, an interpretation of the results can also be offered in terms of dynamic attractor landscapes depicted in Fig 3. Consider a landscape with two attractors, (*i*) a behavioral state of "low physical activity level' (on the left) and (*ii*) a behavioral state of "high physical activity level" (on the right). In panel A of Fig 3, the system is in a stable state, either in the low-activity attractor or the high-activity one (upper panel of Fig 1). In panel B, as the landscape evolves to

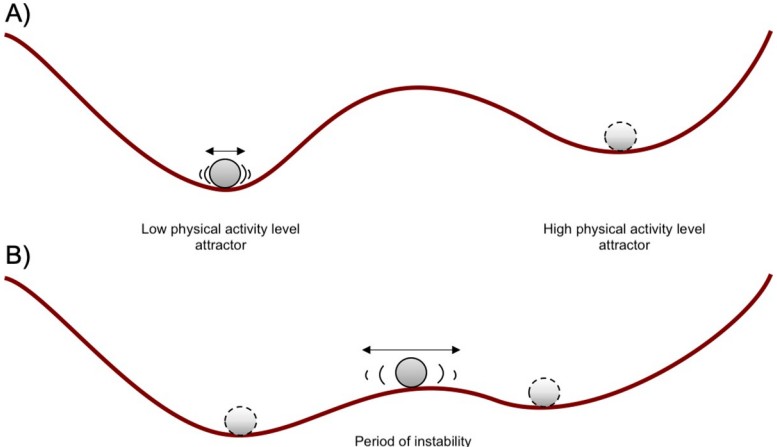

**Fig 3. Attractor landscapes.**

a state where the system is more unstable (due to the intervention, for example), random perturbations from daily events could move it to back and forth either direction, resulting in the instability indicated by increased dynamic complexity. Whether one starts out from low, medium or high activity, if the high-activity attractor remains shallower than the low-activity attractor, instability (which, again, shows up as dynamic complexity) will increase odds of ending up in the low-activity attractor after destabilizations. If the landscape subsequently evolves back to the one depicted in panel A, it has likely become trapped in the low-activity attractor–that is, having experienced what we define as a sudden loss. In other words, we make here the hypothesis that the low activity state would, in general, be more stable than the high activity one in the studied population and that destabilizations therefore are more likely to lead to low activity state (i.e., a sudden loss).

## Study strengths, limitations and perspectives

A key strength of this study lies in its design and modelling approach, which highlighted original and unique information on the within-person, day-to-day changes in walking behavior over several months; which, taking together, lend empirical support toward the adoption of a more dynamic and complex systems-oriented research paradigm in this field of research. This study also provides a first proof of concept regarding the analysis of critical fluctuations in walking behavior as a form of early warning signal for sudden behavioral losses for walking behavior, thus providing potentially fruitful interventional perspectives for the development of just-in-time adaptive interventions.

This study has also several limitations. Based on literature in psychotherapy (e.g., [35]) and the availability of appropriate statistical methods, we decided to classify behavioral changes as sudden gains and losses, however, other characterizations of behavioral states could have been deployed. For example, analyses of decreasing or increasing slopes could be conducted with structural break detection methods [56]. Further, and although our recursive algorithm partially considers within-person variability in daily step count, other statistical methods might be more appropriate in the detection of "stable" *versus* "unstable" behavioral states, such as the analyses of signal variability [57]. Such complementary analyses (e.g., slope or signal variability analyses) could give a different insight on physical activity changes and might actually provide better statistical fits in some contexts or for some participants. Further, and using these or different techniques, it would also be important to explore whether different trajectories or

patterns of change are associated with treatment responses and relevant outcomes (e.g., weight loss, improvements in cardiorespiratory fitness or mental health). In the same vein, we decided in the present study to focus on day-to-day changes in physical activity; however, micro-temporal changes occurring within days (see [58]) or macro-temporal variations across months or years could also be studied with similar methods, with important implications for physical activity theories and interventions from a multi-time scale perspective [59]. Future studies are thus needed to explore different characterizations of discontinuous and idiographic behavior changes over multiple time-scales to complement the results from this early work.

Second, this study used a specific score of local dynamic complexity to characterize critical fluctuations in daily steps based on previous successful applications of this indicator in the prediction of sudden mental health improvements or deteriorations in psychotherapy [35]. Also, and like for the detection of sudden behavioral shifts, other forms of fluctuation indicators could be tested for their predictive power in future research. A previous study for example has computed the local dynamic complexity score from several self-reported items compiled together instead of computing the dynamic complexity score from the main variable of interest [35], like we did in the present study (i.e., steps). Hence, future studies might wish to test whether critical fluctuations in several indicators, compiled together, such as walking behavior but also some of its determinants (e.g., motivational factors, stress, weather), have greater predictive power in the detection of subsequent behavior changes than the analyses of critical fluctuations in the main variable of interest solely. Testing whether determinants of walking behavior (e.g., fluctuations in behavioral intentions or action control) are associated with the critical fluctuations in the behavior would also provide important information about why increases in dynamic complexity are observed and what they represent. Adopting an idiographic approach, we could also make the assumptions that the predictive power of critical fluctuations for behavior changes differ from one person to another [60, 61]. It is possible that different fluctuation indicators (e.g., derived from walking behavior only *versus* from compiled variables) or time-lapses (e.g., 3-day versus 7-day windows to compute local dynamic complexity) are more or less predictive depending on participant idiosyncrasies.

Third, our results provide support for the hypothesis that behavioral change may occur through critical transitions that are preceded by increased levels of dynamic complexity. However, we do not know the causal drivers of the increased dynamic complexity and the transitions. it is important for future studies to explore (*i*) whether possible causal influences on physical activity behavior (i.e., weather variations; changes in motivation) are related to changes in local dynamic complexity and sudden gains or losses. Furthermore, future research should test whether local dynamic complexity can be manipulated during an intervention to prevent sudden losses in walking behavior.

Finally, this study was conducted among young, fairly active, adults with overweight and obesity participating in a weight loss program where different health-related behaviors are manipulated as part of the intervention. Although, this population is of prior interest for physical activity promotion, other pattern of results might be observed in other contexts and populations. Further, as the trial is still ongoing, group assignment remained unknown for the present study, and was not modelled in our analyses, which constitutes a limit given that results might differ by groups. For all these different reasons, results from the present study should be replicated in other contexts.

## Theoretical, methodological and interventional implications

Although results from this study should be replicated in different contexts, they provide empirical evidence that high-resolution changes in walking behavior might be better studied and

understood if elements of complexity theory and methods are accounted for [8]. In other words, a dynamic and complex systems-oriented paradigm should be adopted to better understand and describe physical activity behavior change, notably if those changes are studied at high-resolution, such as a daily level. Theories and models could progressively be refined to include more precise assumptions about time and behavior dynamics as argued in recent position papers [6, 7]. Previous assumptions about gradual behavior change following adoption and maintenance phases should also be refined, or at least completed, to better account for rapidly changing critical transitions. At the methodological level, results from this study suggest that future studies interested in the dynamics of behavior changes should transition from (*i*) low- to high-resolution behavioral assessments to better capture potential discontinuous changes in those processes; (*ii*) group-only to group- and individual-level statistical inference to accurately modelled individual variability; and (*iii*) static to adaptive and continuous tuning interventional designs to better account for rapidly changing behavioral states and contexts (see [9] for further justifications). If successfully applied, that type of adaptive interventions would also argue against the utilization of traditional randomized control trials to test interventions in the physical activity context, at least as they are often implemented, i.e., with low-resolution behavioral measures, exclusively using a nomothetic approach and fixed intervention components.

## Conclusion

This study highlights discontinuous -sudden gains and losses- and heterogeneous patterns of day-to-day changes in walking behavior over months among a large sample of young adults with obesity participating to a weight loss program. This study also provides first evidence that critical fluctuations in behavior time-series can be associated with the subsequent occurrence of behavioral losses in the next days, thus providing a form of early warning signals for physical activity behavior changes. Taken together, this study suggests that walking behavior changes could be well understood under dynamic and complex-system paradigms.

## Supporting information

**S1 Fig. Idiographic visualizations for the 151 participants' time series.**
(PDF)

**S1 Table.** Sensitivity analyses for varying a) thresholds for the identification of gains and losses, b) time-lapses for the computation of local dynamic complexity and c) gains and losses separately and together.
(DOCX)

## Acknowledgments

We would like to thanks the Dr. Damian G Kelty-Stephen for his insights on this project. We would also like to thank all of the staff and participants who are making trial NCT03907462 possible.

## Author Contributions

**Conceptualization:** Guillaume Chevance, Dario Baretta, Matti Heino.

**Data curation:** Guillaume Chevance, Dario Baretta.

**Formal analysis:** Guillaume Chevance, Dario Baretta, Matti Heino, Merlijn Olthof.

**Funding acquisition:** Job Godino.

**Methodology:** Merlijn Olthof.

**Writing – original draft:** Guillaume Chevance, Dario Baretta.

**Writing – review & editing:** Matti Heino, Olga Perski, Merlijn Olthof, Predrag Klasnja, Eric Hekler, Job Godino.

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
