## [Decision Letter · Decision Letter 0]

18 Jan 2021

PONE-D-20-36553

Characterizing and predicting person-specific, day-to-day, fluctuationsin walking behavior

PLOS ONE

Dear Dr. Chevance,

Thank you for submitting your manuscript to PLOS ONE. After careful consideration, we feel that it has merit but does not fully meet PLOS ONE’s publication criteria as it currently stands. Therefore, we invite you to submit a revised version of the manuscript that addresses the points raised during the review process.

We look forward to receiving your revised manuscript.

Kind regards,

Yih-Kuen Jan, PhD, University of Illinois at Urbana-Champaign

Journal Requirements:

We note that one or more of the authors are employed by a commercial company: Accenture.

2.1. Please provide an amended Funding Statement declaring this commercial affiliation, as well as a statement regarding the Role of Funders in your study. If the funding organization did not play a role in the study design, data collection and analysis, decision to publish, or preparation of the manuscript and only provided financial support in the form of authors' salaries and/or research materials, please review your statements relating to the author contributions, and ensure you have specifically and accurately indicated the role(s) that these authors had in your study. You can update author roles in the Author Contributions section of the online submission form.

2.2. Please also provide an updated Competing Interests Statement declaring this commercial affiliation along with any other relevant declarations relating to employment, consultancy, patents, products in development, or marketed products, etc.  

Reviewers' comments:

Reviewer's Responses to Questions

**Comments to the Author**

1. Is the manuscript technically sound, and do the data support the conclusions?

Reviewer #1: Yes

Reviewer #2: Partly

Reviewer #3: Yes

2. Has the statistical analysis been performed appropriately and rigorously? 

Reviewer #1: Yes

Reviewer #2: I Don't Know

Reviewer #3: Yes

3. Have the authors made all data underlying the findings in their manuscript fully available?

Reviewer #1: Yes

Reviewer #2: Yes

Reviewer #3: Yes

4. Is the manuscript presented in an intelligible fashion and written in standard English?

Reviewer #1: Yes

Reviewer #2: Yes

Reviewer #3: Yes

5. Review Comments to the Author

Reviewer #1: The authors intended to assess walking behavior under a dynamic paradigm. They analyzed data based on 151 overweight or obesity young adults. They conduct the analysis using recursive partitioning algorithm to identify the patterns of behavioral change and computed individual’s fluctuation intensity with a dynamic complexity algorithm. The results showed the positive association between critical fluctuations in walking behavior, a form of early warning signals, and the subsequent occurrence.

1. Line 230. The regression tree was conducted based on several binary criteria. What are these binary criteria? How about other continuous sample characters, such as age and so on? Also, the outcome is binary, classification tree is more appropriate name for the analysis.

2. Classification/regression tree tends to be overfitting in general. Please comment on addressing this for the current analysis.

3. Classification/regression tree is pretty data-driven approach. How stable of the results would be for current analysis? or any strategies were implemented to avoid this?

Reviewer #2: In the “Data analysis” section, please adress the following concerns.

(1) What does the sentence “Missing days were imputed using the Kalman Filter method” (lines 219-220) mean? Please describe the approach briefly.

(2) The approach to identify sudden gains and losses for depressive symptoms (lines 225-230) was unclear.

(3) In the subsection “Local dynamic complexity calculation”, it would be better to give the mathematical expressions of the metrics such as “dynamic complexity score”, “fluctuation measure”, and “distribution parameter” and then explain the rationales for the use of these metrics.

(4) In the subsection “Associations between …”, it is difficult to understand the method to test the association between local dynamic complexity and the occurrence of sudden gains and losses (lines 278-283) and the equation (lines 286-287). It would be better to present them in a mathematical manner.

Reviewer #3: In this submission, the authors are able to show a correlation between fluctuations in walking behavior and subsequent behavioral losses within the following days. The primary methodology is based on a reasonable sample size based on a recent parallel-group randomized controlled trial related to weight loss among young adults. A significant limitation of the study is that the trial is ongoing, in addition to changes due to to the onset of Covid-19 disturbance (which the authors account for), and as such information was not available on group assignment. Due to the nature of the aforementioned RCT study, there could be different interventions on subgroups of the study population which could affect the data. This would have been interesting to analyze or detect subgroups of varying complexity within the analyzed population, even though it is understandable there would be no confirmation available due to the RCT design parameters. Another limitation is the age range of the study population. Overall, the authors address the above points and do a good job in describing the limitations of their study.

Regarding methodology, the authors performed Kalman filtering over missing days, and specified a mean number of days missing and range. To detect gains and losses, a recursive partitioning algorithm available in an R package was used. Changes of 30% increase or reduction of steps were assessed. A sensitivity analysis was performed looking at the effect of changing this threshold. A different R package was used to calculate dynamic complexity. A link was provided for data and code. Looking at associations between gains/losses and local dynamic complexity showed relations as characterized in Table 1, with appropriate statistics. Overall, the approach is sound and the results are interesting. It would be good to know the proportion of stable 7 day periods vs periods of high variability to better characterize the amount of days/time during which no sudden gains/losses could be computed. Figure 1 could also use expansion, to better illustrate use of the algorithm and graphically display examples of some of the situations encountered (gain vs loss, threshold bars, etc). Also recommend the time axis be labeled with units.

The author's discussion was sound and noted results that somewhat but not entirely corresponded with literature. Results are interpreted, and limitations of the study were described in a reasonable amount of detail. Fig.3 might have benefitted from more labeling or description in the figure caption (which was embedded in the text instead). As the authors mention in their study strengths and limitations section, it would be interesting to cross compare with different scores of local dynamic complexity, particularly in reference to variables outside the main variables of interest, and taking into account timing of events such as interventions among different subgroups in the RCT. This would be an interesting topic of follow-up work. Overall, the manuscript is well written, the methodology described and thought processes used in the study are thoroughly described, and the results are interesting, though require further study in the future. I thus recommend this manuscript for publication.

6. PLOS authors have the option to publish the peer review history of their article (what does this mean?). If published, this will include your full peer review and any attached files.

Reviewer #1: No

Reviewer #2: No

Reviewer #3: No

---

## [Author Response · Author response to Decision Letter 0]

1 Apr 2021

PONE-D-20-36553

Characterizing and predicting person-specific, day-to-day, fluctuations in walking behavior

PLOS ONE

Yih-Kuen Jan, PhD, University of Illinois at Urbana-Champaign

 Dear Editor, 

 Thank you for the opportunity to submit a revised version of our manuscript. I would like to draw your attention to the fact that, during the revision process, we benefited from feedback from Merlijn Olthof (PhD candidate from Radboud University and topic specialist) on our study’s preprint. Given the quality of his feedback, we offered Dr. Olthof to be part of the authorship for this paper. One of the main remarks made by Dr. Olthof was to include a “duration” indicator in our statistical model accounting for the time (i.e., number of days) between each event (i.e., gains and losses), as usually recommended for this type of analysis. This addition substantially improved our models’ fit indices and slightly changed the results. 

All the modifications made in regards of Dr. Olthof’s comments and the ones of the three reviewers solicited by PLOS ONE can be tracked in the manuscript. Answers to the 3 reviewers’ remarks and your specific editorial interrogations can be found below.

Best regards. 

Guillaume Chevance

 

Editor’s comments:

Authors’ answer: Modifications have now been made to fit with PLOS ONE’s style requirements. 

2. Thank you for stating the following in the Competing Interests section: "The authors have declared that no competing interests exist."

We note that one or more of the authors are employed by a commercial company: Accenture.

Authors’ answer: A competing interests statement has been added on page 29. None of the authors have any competing interests to declare. The activity of Dr. Baretta as part of Accenture isn’t related to the present manuscript. Dr. Baretta was affiliated with the University of California at San Diego (as a post-doctoral scholar) at the beginning of this project and has worked on the present article in his free time. There are no links, or any competing interests, between his affiliation/work with Accenture and the present study. 

2.1. Please provide an amended Funding Statement declaring this commercial affiliation, as well as a statement regarding the Role of Funders in your study. If the funding organization did not play a role in the study design, data collection and analysis, decision to publish, or preparation of the manuscript and only provided financial support in the form of authors' salaries and/or research materials, please review your statements relating to the author contributions, and ensure you have specifically and accurately indicated the role(s) that these authors had in your study. You can update author roles in the Author Contributions section of the online submission form.

 Authors’ answer: The Funding Statement has now been updated.

2.2. Please also provide an updated Competing Interests Statement declaring this commercial affiliation along with any other relevant declarations relating to employment, consultancy, patents, products in development, or marketed products, etc. 

 Authors’ answer: This was updated.

Authors’ answer: This was updated.

 

Reviewers' comments:

Reviewer #1: 

1. Line 230. The regression tree was conducted based on several binary criteria. What are these binary criteria? How about other continuous sample characters, such as age and so on? Also, the outcome is binary, classification tree is more appropriate name for the analysis.

Authors’ answer: We thank the reviewer for this comment because it gave us the opportunity to expand more on this important aspect of our analysis. We confirm that we adopted regression trees (the dependent variable of the partitioning algorithm is continuous) instead of classification tree and have amended this section accordingly, which now reads (page 12 in “characterization of gains and losses”): “Following the approach applied by Olthof et al. (2020), we used a recursive partitioning algorithm to characterize within-person behavioral changes over the course of the study using regression trees from the package rpart[44] for the R software environment[45]. Regression trees enable to identify nodes and partition a continuous variable y in recursive splits based on different predictors. In the present study, we partitioned the variable steps based on the progressive day of the intervention as predictor. Once we identified the splits for the variable steps for each participant separately, we classified them as gain or losses if they met additional criteria defined on the basis of previous research […].”

Please note that this characterization is done at the individual level (i.e., for each participant separately) and focuses on the behavior (i.e., as a way to describe the behavioral change). Inter-individual-differences, such as age, are thus not relevant for this person-specific approach given that participants are not pooled together for these specific analyses.

2. Classification/regression tree tends to be overfitting in general. Please comment on addressing this for the current analysis.

Authors’ answer: We agree that regression trees tend to be overfitting. However, in the current study, the regression tree analysis was performed to identify splits in the time series. Such splits were successively coded as gains or losses if and only if they met more restrictive criteria driven by previous research. The criteria are explicitly mentioned on pages 12 and 13. The first is the 30% increase or reduction in steps relative to the median number of steps for each participant (point 1, please see line 254). The second is related to the period of 7 days during which a “stable change” can be characterized (point 2, please see line 264). Therefore, the number of shifts identified for each participant are a subset of the splits/nodes resulting from the regression tree. In this way we managed to counteract the potential overfitting of the regression tree models (with the application of complementary criteria).

3. Classification/regression tree is pretty data-driven approach. How stable of the results would be for current analysis? or any strategies were implemented to avoid this?

Authors’ answer: We agree that regression trees are a data driven approach, however, as highlighted in our response to the previous point, we counterbalanced this with specific top-down or theory-driven criteria based on our past research and the literature. Moreover, in order to avoid the opposite error and being too theory-driven, we performed multiple sensitivity analyses in order to test the robustness of the results under different assumptions and to adequately balance data and theory driven approaches. Specifically, sensitivity analyses were conducted for varying cut-offs (e.g., sudden gains or losses of 20% and 40%). This is explained in the “sensitivity analyses” section (in the results): 

“Sensitivity analyses were conducted to test the robustness of the results for varying levels of gains and losses (i.e., +/-20% and +/-40% of the median number of steps, instead of +/-30%, as specified in the main analyses), and different time-lags for the local dynamic complexity score (i.e., higher scores in the past two and four days, instead of the past three days, as specified in the main analyses). Results of the sensitivity analyses were compared with regards to their effects on the identification of gains and losses (objective 1) and the association between local dynamic complexity and the occurrence of sudden gains and losses (objective 2). 

As shown in Table 2, lower (+/-20%) and higher (+/-40%) thresholds lead respectively to the identification of more (N = 1180) and less (N = 595) sudden gains and losses in comparison with the main analyses (+/-30%; N = 862). Using a more conservative threshold (+/-40% of the median number of steps) resulted in the detection of a lower number of sudden gains and losses, while using a less conservative threshold (+/-20% of the median number of steps) resulted in the detection of a higher number of events.”

Please see also the s1 table in supplemental material. The associations between local dynamic complexity and the occurrence of sudden gains and losses were substantially unchanged (notably the prediction of sudden losses) in the sensitivity analyses

 

Reviewer #2: In the “Data analysis” section, please address the following concerns.

(1) What does the sentence “Missing days were imputed using the Kalman Filter method” (lines 219-220) mean? Please describe the approach briefly.

Authors’ answer: Thank you for your feedback. We have now provided more detail on this method on pages 11/12 and provided an additional reference for interested readers: “Missing days were imputed using the Kalman Filter method (i.e., a procedure to compute the likelihood of a time series which is the outcome of a stationary autoregressive moving average process or non-stationary autoregressive integrated moving average process; see for further details [42]).”

We added this reference:

[42] Gómez V, Maravall A. Estimation, Prediction, and Interpolation for Nonstationary Series with the Kalman Filter. J Am Stat Assoc. 1994 Jun 1;89(426):611–24.

(2) The approach to identify sudden gains and losses for depressive symptoms (lines 225-230) was unclear.

Authors’ answer: We agree that the reference to the previous study about depressive symptoms was unclear and have now modified these two paragraphs to make the points more explicit (and answer other comments from the reviewer 1 as well). Please, see our modifications on pages 12-13 in the “characterization of gains and losses” section starting line 248. 

(3) In the subsection “Local dynamic complexity calculation”, it would be better to give the mathematical expressions of the metrics such as “dynamic complexity score”, “fluctuation measure”, and “distribution parameter” and then explain the rationales for the use of these metrics.

Authors’ answer: We now provide a technical document with all the mathematical details related to this score (please, see the document entitled “Dynamic Complexity” in supplemental material available at https://osf.io/64cmv/ and page 14). We contend that presenting these indicators in non-mathematical terms within the main manuscript is a better strategy to keep the manuscript intelligible for most readers, notably our targeted audience (interventionists in behavioral medicine). Readers interested, and able, to deeply understand the maths behind this algorithm can now consult our appendix in addition to the reference indicated in the manuscript (references #34 and #50 in the list). Additionally, for interested readers, the R code contain the operationalization of the formula to compute dynamic complexity. We explicitly mention this addition line 314-315: “For mathematical details, see the document entitled “Dynamic Complexity” in the supplemental material available at https://osf.io/64cmv/).”

(4) In the subsection “Associations between …”, it is difficult to understand the method to test the association between local dynamic complexity and the occurrence of sudden gains and losses (lines 278-283) and the equation (lines 286-287). It would be better to present them in a mathematical manner.

Authors’ answer: Although we understand your comment and truly respect your position, we believe that for most applied researchers (like us) it’s more common and easy to represent, read and understand a statistical model in the way it is implemented in traditional statistical software. Even if the use of these models (here generalized linear mixed-effects model) is extremely common in both social (e.g., psychology) or biological sciences (e.g., epidemiology), most of us aren’t mastering the mathematical equations underlying such type of models. Readers interested in these details can consult the references we provided for the theory behind the models (reference #51) and its implementation in R (reference #52).

 

Reviewer #3: In this submission, the authors are able to show a correlation between fluctuations in walking behavior and subsequent behavioral losses within the following days. The primary methodology is based on a reasonable sample size based on a recent parallel-group randomized controlled trial related to weight loss among young adults. A significant limitation of the study is that the trial is ongoing, in addition to changes due to the onset of Covid-19 disturbance (which the authors account for), and as such information was not available on group assignment. Due to the nature of the aforementioned RCT study, there could be different interventions on subgroups of the study population which could affect the data. This would have been interesting to analyze or detect subgroups of varying complexity within the analyzed population, even though it is understandable there would be no confirmation available due to the RCT design parameters. Another limitation is the age range of the study population. Overall, the authors address the above points and do a good job in describing the limitations of their study. Regarding methodology, the authors performed Kalman filtering over missing days, and specified a mean number of days missing and range. To detect gains and losses, a recursive partitioning algorithm available in an R package was used. Changes of 30% increase or reduction of steps were assessed. A sensitivity analysis was performed looking at the effect of changing this threshold. A different R package was used to calculate dynamic complexity. A link was provided for data and code. Looking at associations between gains/losses and local dynamic complexity showed relations as characterized in Table 1, with appropriate statistics. Overall, the approach is sound and the results are interesting. 

Authors’ answer: Thank you for your positive feedbacks.

R3. It would be good to know the proportion of stable 7-day periods vs periods of high variability to better characterize the amount of days/time during which no sudden gains/losses could be computed. 

Authors’ answer: As per our algorithm, any shift can't be preceded or followed by any other shift during the 7- days window which precedes and follows each shift. Overall, for each participant, the number of stable periods (i.e., period between two shifts) is equal to number of shifts + 1. 

R3. Figure 1 could also use expansion, to better illustrate use of the algorithm and graphically display examples of some of the situations encountered (gain vs loss, threshold bars, etc). Also recommend the time axis be labeled with units.

Authors’ answer: It’s interesting to discuss the unit of x axis (time). Also, because this approach can be used for rapid- or slow-evolving processes, we decided to not modify the x axis of the figure. However, this point is now justified in the Figure legend: “Note that the Figure’s x axis (time) is not expressed here with a specific unit; this approach, however, can be used to describe both rapid- and slow-evolving processes. The time unit here can thus conceptually vary from seconds to days, months or years”. 

It’s difficult to provide indications related to the algorithm on this figure, because this last appears early in the manuscript as a conceptual, rather than technical, description of the approach (that is then develop in the method). However, we now provide a new supplemental file describing the approach in more technical terms (in the method section). Please, see page 14 and the supplemental material entitled “Dynamic Complexity” (available at https://osf.io/64cmv/). 

R3. The author's discussion was sound and noted results that somewhat but not entirely corresponded with literature. Results are interpreted, and limitations of the study were described in a reasonable amount of detail. Fig.3 might have benefitted from more labeling or description in the figure caption (which was embedded in the text instead). 

Authors’ answer: Thanks for your comment. We modified and added information in the Figure 3 to make it more intelligible (please see the figure as displayed by PLOS ONE). 

R3. As the authors mention in their study strengths and limitations section, it would be interesting to cross compare with different scores of local dynamic complexity, particularly in reference to variables outside the main variables of interest, and taking into account timing of events such as interventions among different subgroups in the RCT. This would be an interesting topic of follow-up work. Overall, the manuscript is well written, the methodology described and thought processes used in the study are thoroughly described, and the results are interesting, though require further study in the future. I thus recommend this manuscript for publication.

Authors’ answer: Thank you again for the positive feedbacks. We agree with the seminal aspect of our work, as well as the limitation related to the unknown group assignment. We added two sentences (pages 27-28 at the end of the discussion section before the “implications section”) explicitly mentioning the needs for replications of the present results: “Further, as the trial is still ongoing, group assignment remained unknown for the present study, and was not modelled in our analyses, which constitutes a limit given that results might differ by groups. For all these different reasons, results from the present study should be replicated in other contexts.”

---

## [Decision Letter · Decision Letter 1]

30 Apr 2021

Characterizing and predicting person-specific, day-to-day, fluctuations in walking behavior

PONE-D-20-36553R1

Dear Dr. Chevance,

We’re pleased to inform you that your manuscript has been judged scientifically suitable for publication and will be formally accepted for publication once it meets all outstanding technical requirements.

Kind regards,

Yih-Kuen Jan, PhD, University of Illinois at Urbana-Champaign

Additional Editor Comments (optional):

Reviewers' comments:

Reviewer's Responses to Questions

**Comments to the Author**

1. If the authors have adequately addressed your comments raised in a previous round of review and you feel that this manuscript is now acceptable for publication, you may indicate that here to bypass the “Comments to the Author” section, enter your conflict of interest statement in the “Confidential to Editor” section, and submit your "Accept" recommendation.

Reviewer #1: All comments have been addressed

2. Is the manuscript technically sound, and do the data support the conclusions?

Reviewer #1: (No Response)

3. Has the statistical analysis been performed appropriately and rigorously? 

Reviewer #1: (No Response)

4. Have the authors made all data underlying the findings in their manuscript fully available?

Reviewer #1: (No Response)

5. Is the manuscript presented in an intelligible fashion and written in standard English?

Reviewer #1: (No Response)

6. Review Comments to the Author

Reviewer #1: (No Response)

7. PLOS authors have the option to publish the peer review history of their article (what does this mean?). If published, this will include your full peer review and any attached files.

Reviewer #1: No

---

## [Editor Report · Acceptance letter]

4 May 2021

PONE-D-20-36553R1 

Characterizing and predicting person-specific, day-to-day, fluctuations in walking behavior 

Dear Dr. Chevance:

I'm pleased to inform you that your manuscript has been deemed suitable for publication in PLOS ONE. Congratulations! Your manuscript is now with our production department. 

Kind regards, 

on behalf of

Dr. Yih-Kuen Jan 

Academic Editor

PLOS ONE